# Molecular Mechanisms Underlying Vascular Liver Diseases: Focus on Thrombosis

**DOI:** 10.3390/ijms241612754

**Published:** 2023-08-13

**Authors:** Lucia Giuli, Maria Pallozzi, Giulia Venturini, Antonio Gasbarrini, Francesca Romana Ponziani, Francesco Santopaolo

**Affiliations:** 1Hepatology Unit, CEMAD Centro Malattie Dell’Apparato Digerente, Medicina Interna e Gastroenterologia, Fondazione Policlinico Universitario Gemelli IRCCS, 00168 Rome, Italy; lucia.giuli92@gmail.com (L.G.); mariapallozziucsc@gmail.com (M.P.); giulia.ventu97@gmail.com (G.V.); francesca.ponziani@gmail.com (F.R.P.); santopaolofrancesco@gmail.com (F.S.); 2Dipartimento di Medicina e Chirurgia Traslazionale, Università Cattolica del Sacro Cuore, 00168 Rome, Italy

**Keywords:** anticoagulants, budd chiari syndrome, liver cirrhosis, parenchymal extinction, portal vein thrombosis, porto-sinusoidal vascular disorder, thrombosis, vascular disorder of the liver

## Abstract

Vascular liver disorders (VLDs) comprise a wide spectrum of clinical-pathological entities that primarily affect the hepatic vascular system of both cirrhotic and non-cirrhotic patients. VLDs more frequently involve the portal and the hepatic veins, as well as liver sinusoids, resulting in an imbalance of liver homeostasis with serious consequences, such as the development of portal hypertension and liver fibrosis. Surprisingly, many VLDs are characterized by a prothrombotic phenotype. The molecular mechanisms that cause thrombosis in VLD are only partially explained by the alteration in the Virchow’s triad (hypercoagulability, blood stasis, and endothelial damage) and nowadays their pathogenesis is incompletely described and understood. Studies about this topic have been hampered by the low incidence of VLDs in the general population and by the absence of suitable animal models. Recently, the role of coagulation imbalance in liver disease has been postulated as one of the main mechanisms linked to fibrogenesis, so a novel interest in vascular alterations of the liver has been renewed. This review provides a detailed analysis of the current knowledge of molecular mechanisms of VLD. We also focus on the promising role of anticoagulation as a strategy to prevent liver complications and to improve the outcome of these patients.

## 1. Introduction

Vascular disorders of the liver (VLDs) are a rare condition that affect less than 5/10,000 and comprise a series of different conditions with high morbidity and mortality. They include a series of inherited or acquired malformation of the vessels that may interest the portal venous system, the hepatic veins, and liver sinusoids that could be detected both in non-cirrhotic and in cirrhotic patients [1]. Little is known about natural history and the mechanisms that lead to these disorders, which are unique and partially understood. In this review we describe the main vascular disorders of the liver, with a special interest on their etiopathogenesis and the molecular mechanisms that influence their development. We briefly report the current knowledge on the therapeutic management of these diseases with a special focus on the use of anticoagulation.

## 2. Budd Chiari Syndrome

Budd Chiari Syndrome (BCS) is a rare liver disorder characterized by the obstruction of the hepatic venous outflow that can occur at different levels, from the small centrilobular veins to the hepatic veins until the inferior cava vein and the right atrium junction [2].

This definition excludes all the forms of hepatic outflow obstruction caused by cardiac failure disease, constrictive pericarditis, and sinusoidal obstruction syndrome (SOS) [3]. In the primary form of BCS obstruction results from a pathologic process involving the veins as thrombosis or phlebitis; meanwhile, secondary BCS is caused by the extrinsic compression or invasion of the hepatic venous outflow by a mass located outside the veins (benign or malign tumor, abscesses, lymphadenopathy, parasitic, or non-parasitic cyst) [4]. Clinical presentation of BCS ranges from asymptomatic cases to fulminant liver failure depending on the degree and the rapidity of hepatic vein obstruction. Most of the patients present the classical triad of abdominal pain, ascites, and hepatomegaly [5]. About 75% of patients with BCS have an inherited or acquired hypercoagulability state that predisposes them to thrombosis formation. In 25% of patients, a combination of more than one etiologic factor can coexist [6]. Among the conditions that are associated with the development of BCS it is important to mention acquired disorders (myeloproliferative diseases (MPNs), antiphospholipid syndrome, paroxysmal nocturnal hemoglobinuria), inherited disorders (Factor V Leiden, Factor II gene mutation, deficiencies of antithrombin, proteins S and C), systemic diseases (Behcet’s disease, celiac disease, inflammatory bowel disease), and external factors (oral contraceptive use and pregnancy) [7,8,9]. In patients with BCS alterations in fibrinolysis are also reported as cause of vein thrombosis: higher plasminogen activator inhibitor 1 (PAI-1) levels and lower levels of thrombin activatable fibrinolysis inhibitor (TAFI) and α2-antiplasmin are reported in BSC compared to controls. In one study, a clot lysis time (CLT) above the 90th or 95th percentile of controls was associated with a 2.4-fold or 3.4-fold increase in risk of BCS, respectively [10,11,12]. MPNs, which include polycythemia vera (PV), essential thrombocythemia (ET), and primary myelofibrosis (PMF), are the most frequent cause of BCS and present in about 30–57% and 5–28% of patients in Europe and in Asia, respectively [13]. The development of MPNs is related to the acquisition of somatic genetic alterations with Janus kinase 2 (JAK2) V617F mutation being the most common in BCS showing a prevalence of this mutation in 41% [3,14,15]. Patients with MPNs present an increased risk of arterial and venous thrombosis that can develop in unusual sites such as splanchnic veins and cerebral sinuses [16]. In general, venous thrombosis arises from a combination of hypercoagulability and stasis. Specifically, endothelial hypoxia resulting from the reduction in blood flow promotes endothelial cells to acquire a pro-inflammatory state. These cells express adhesion molecules, including P selectin which attract and activate leukocytes and platelets, causing their accumulation. Upon activation, neutrophils release neutrophil extracellular traps (NETs) which exert prothrombotic properties [17]. Various mechanisms of these components are involved in the increased risk of thrombotic events in patients with MPNs, including increased expression of P selectin on endothelial cells, NETs formation, and increased leukocyte adhesion [18]. Moreover, especially in patients with JAK2V617F mutation related PV, hyperviscosity induced by high hematocrit slows down blood flow, favoring endothelial cells’ activation and subsequently thrombosis. The presence of JAK2V617F on red blood cells enhances their adhesion to endothelial cells through the interaction between the overexpressed membrane erythrocyte protein erythroid Lutheran/basal cell adhesion molecule (Lu/BCAM) with endothelial laminin [19]. Several studies have documented the presence of JAK2V617F mutation in the endothelial cells (ECs) of hepatic veins in BCS patients as well as in circulating endothelial progenitor cells [20,21]. Guy et al. in an animal model demonstrated that ECs’ expression of JAK2V617F mutation promoted a pro-thrombotic state by inducing P selectin exposure with a subsequent increase in platelets and leukocyte adhesion. Treatment with hydroxyurea brought a reduction in P selectin expression on these ECs and consequently of thrombus formation [22]. In another study, JAK2V617F and JAK2V617F-wild type induced pluripotent stem cells from a patient with MPN were redirected towards endothelial lineage. The JAK2V617F endothelial cells displayed a proinflammatory and prothrombotic phenotype, with an increased expression of von Willebrand factor (vWF) and P selectin. Moreover, leukocytes from MPNs patients showed an increased adhesion to these JAK2V617F ECs [19]. Irrespective of the cause, hepatic venous outflow obstruction is responsible for hepatic venous stasis resulting in congestive hepatopathy and an increased sinusoidal pressure which is the main driver of ascites formation and portal hypertension [23]. Congestive hepatopathy leads to sinusoidal thrombosis which causes hepatocytes ischemia and necrosis which in turn promotes perivenular and perisinusoidal fibrotic deposition in the central part of the hepatic lobule [24]. In chronic BCS, the development of hepatic venous collaterals and the increase in arterial perfusion represent a compensatory mechanism towards hepatic congestion and hepatocellular ischemia, slowing the progression of the disease [6].

In hyper arterialized liver areas, where there is an imbalance between arterial and portal inflow, regenerative macronodules can arise which can mimic focal nodular hyperplasia [25]. Long-term BCS can be also associated with the development of hepatocellular carcinoma (HCC), with a 10-year cumulative incidence of 10% [6]. Although the exact pathogenesis of HCC in BCS patients is still not clear, chronic hepatic congestion, increased arterial blood flow and inferior vena cava obstruction seem to be the main driver of its development [26,27]. Based on these considerations, the improvement of hepatic sinusoidal pressure by therapeutic interventions or the development of venous collaterals is critical to reduce further progression of fibrosis, nodular regeneration and eventually the onset of cirrhosis [28]. In these patients, a stepwise treatment strategy based on the response to prior therapy is recommended. Long-term anticoagulation together with treatment of the underlying prothrombotic disorder represents the first therapeutic approach [13,29,30,31]. Short-length venous stenosis should be individualized and treated with angioplasty [32]. In those with incomplete response, transjugular intrahepatic portosystemic shunt (TIPS) and, finally, liver transplantation (LT) should be performed [33].

## 3. Porto-Sinusoidal Vascular Disorder

Porto-sinusoidal vascular disorder (PSVD) is a rare vascular disease of the liver characterized by specific histological features involving the small portal venules and the sinusoids, associated or not with portal hypertension, in the absence of cirrhosis on liver biopsy [34]. Previously known as idiopathic non-cirrhotic portal hypertension (INCPH), in 2017 the VALDIG group coined the name of PSVD in order to also include patients with typical histological characteristics but without portal hypertension under this definition. Moreover, this new definition does not exclude patients with portal vein thrombosis or other causes of liver disease if the liver biopsy is suggestive for PSVD [35]. Patients with PSVD without portal hypertension may present only with mild alteration in liver tests such as transaminase, alkaline phosphatase, and gamma-glutamyltransferase [36]. Also, patients with PSVD-associated portal hypertension are generally asymptomatic unless they present a complication typical of portal hypertension. Indeed, in 20–40% of cases, variceal bleeding represents the first symptomatic event. During the natural history of the disease, about 20–50% of patients develop ascites and portal vein thrombosis will be present in 30–40% of cases at 5 years from the diagnosis [37,38,39]. To diagnose PSVD liver biopsy is mandatory, both to exclude the presence of cirrhosis and to identify specific or non-specific histological characteristics related to this condition. Specific histological lesions suggestive for PSVD comprise obliterative portal venopathy recently renamed portal vein stenosis, incomplete septal fibrosis, and nodular regenerative hyperplasia [34]. Portal vein stenosis refers to a lesion characterized by a pathological narrowing of the portal vein branch varying from partial to complete obliteration of the vessel lumen, presumably leading to an increased intra-hepatic resistance [35,40]. Nodular regenerative hyperplasia (NRH) is defined by benign replacement of the liver parenchyma with multiple micronodules, typically 1–3 mm in diameter, in the absence of fibrosis [41]. Incomplete septal fibrosis is characterized by thin or incomplete septa that delineate rudimentary nodules [42]. The pathogenesis of PSVD is not well understood yet, although several conditions such as drug exposure, disorder of blood coagulation, autoimmune disease, immune disorders, congenital or hereditary defects, and infections are found in 43–58% of patients. It is thought that these conditions may be potentially responsible for the vascular liver injuries that occur in the small or medium portal venules or sinusoids [34,43]. Drug exposure, especially long-term administration of didanosine and stavudine in the setting of HIV infection, was related to nodular regenerative hyperplasia [44]. Indeed, both older age and cumulative exposure to didanosine and stavudine were found to be independent predictors of NRH formation in HIV patients. These drugs could cause endothelial damage through a pro-inflammatory mechanism involving the small portal vessels [45]. Previous exposure to immunosuppressive or antineoplastic drugs such as azathioprine, oxaliplatin, and 6-thioguanine was also correlated with PSVD [46]. While high levels of 6-thioguanine seem to be associated with the development of NRH, long-term azathioprine administration seems to be responsible for sinusoidal endothelial cells damage via cellular glutathione depletion [47,48]. Three mechanisms seem to be related to PSVD development in patients treated with oxaliplatin. Indeed, oxaliplatin administration may cause sinusoidal damage by inducing an increment in sinusoidal endothelial cells porosity and cellular fenestrations, thereby favoring the release of free radicals and glutathione transferase depletion. This allows for erythrocytes migration into the space of Disse, leading to perisinusoidal fibrosis. In addition, oxaliplatin may induce NRH formation through chronic hypoxia in the centrilobular area. Lastly, this drug may be responsible for blood capillary obliteration and regions of parenchymal extinction [49]. About 10% of patients with PSVD present with an immunological disorder, such as immune deficiencies or autoimmune diseases [50,51]. Intrasinusoidal T lymphocytes may be involved in the sinusoidal changes that occur in these patients, thus promoting the development of portal vein or sinusoidal endotheliitis. This aligns with an increased expression of genes associated with lymphocytes activation in the blood of PSVD patients [52]. Reyes et al. evaluated the potential role of autoimmunity in PSVD pathophysiology by comparing the presence of autoantibodies between patients with PSVD and cirrhosis. PSVD patients showed a higher proportion of autoantibodies than patients with cirrhosis. Moreover, anti-endothelial-cell antibodies (AECA) were significantly increased in PSVD patients. AECA, which binds to a 68–72 kDa protein belonging to the family of heat shock protein (HSP) expressed on the membrane of liver endothelial sinusoidal cells, may promote ECs apoptosis and so play an important role in PSVD pathogenesis [53]. Prothrombotic conditions can also have a pathogenic role in PSVD pathogenesis. Microthrombi and platelet aggregation seem to be involved in PSVD onset. Indeed, it is thought that obliterative portal venopathy is the result of prior thrombotic events [54]. Moreover, thrombophilia, and especially protein C deficiency, were found to have a high prevalence in these patients [55,56]. Finally, it is known that PVT represents a frequent event during the course of PSVD disease supporting a possible prothrombotic state related to this condition [57]. Also, infections have been considered a trigger for PSVD development. Recurrent intra-abdominal infections, through septic embolization, may lead to small and medium portal venules obstruction [58]. Various animal studies were conducted in order to investigate the role of infections in the development of non-cirrhotic portal hypertension [59]. In a rabbit model, intraportal or intravenous administration of killed non-pathogenic *Escherichia coli* was related to the development of splenomegaly, portal fibrosis, and in some cases marked portal hypertension [60]. PSVD could also have a genetic basis. Turner’s syndrome, Adams-Oliver syndrome, and cystic fibrosis are some of the genetic disorders associated with PSVD [61,62,63]. One study demonstrated familial aggregation between human leukocyte antigen (HLA)-DR3 and PSVD [64]. Correlations have been established between PSVD and mutation in the telomerase genes complex [65]. Indeed, a wide range of hematologic and liver abnormalities have been associated with heterozygous telomerase loss-of-function mutations [65]. Various mutations implicated in the pathogenesis of PSVD have been identified using whole exome sequencing in families affected by this entity [66]. Recently, Shan et al. performed genome sequencing in members of a large multigenerational Lebanese family with PSVD. They identified an aberrant variant of FCH and double SH3 domains 1 (FCHSD1), an uncharacterized gene which segregates with PSVD in an autosomal dominant pattern and leads to mammalian target of rapamycin (mTOR) pathway overactivation, which has a role in platelet activation and thrombus formation [67,68]. Hernandez-Gea et al., in order to uncover the biological mechanisms underlying PSVD development, used a biological network approach. They demonstrated that patients with PSVD have a unique transcriptomic profile with the most expressed genes involved in the regulation of vascular homeostasis but also in lipid metabolism and oxidative phosphorylation such as those belonging to Serpin family, the apolipoproteins, adenosine triphosphate synthases, fibrinogen genes, and alpha-2-macroglobulin. Dysregulation of these pathways may be responsible for intravascular endothelial dysfunction leading to the characteristic liver histological changes observed in PSVD [69]. Management of patients with PSVD is primarily directed towards portal hypertension complications and its prevention. In the presence of varices, therapy with non-selective beta blocker (NSSB) or variceal band-ligation should be performed [66]. TIPS should be taken into consideration in the cases of variceal bleeding or refractory ascites [70]. Since thickening and obliteration of the portal venules is one of the specific histological features of PSVD, and PSVD is frequently associated with prothrombotic conditions and the onset of portal vein thrombosis, prophylactic use of anticoagulants could be useful. However, randomized trials on the role of anticoagulants in this setting are needed to answer this question [34].

## 4. Portal Vein Thrombosis in Cirrhosis

Portal vein thrombosis (PVT) is the occlusion of the portal vein trunk, extended or not, to its intrahepatic branches. It may involve the splenic and the superior mesenteric vein. The vessel may be completely or partially occluded. The onset may be acute or chronic. PVT is a common event in patients with chronic liver disease, while it is a rare site of thrombosis in noncirrhotic patients [1,71,72]. In liver cirrhosis, the prevalence of PVT development varies according to the stage of liver disease and the presence of hepatocellular carcinoma: according to the stage, 10% of patients have PVT in compensated cirrhosis while the prevalence is up to 17% in patients with Child Pugh B/C, with higher percentage in patients awaiting for liver transplantation. The pooled prevalence is about 13.9% [73,74,75,76,77]. The incidence of PVT increases tremendously in patients with cirrhosis and hepatocellular carcinoma (10–40%) [78,79]. According to the Virchow’s triad, hypercoagulability, endothelial injury, and reduced blood flow are the main risk factors for development of thrombosis [80]. In the case of PVT, these mechanisms are consequences of portal hypertension development and maintenance, so portal hypertension is the main player for PVT occurrence in chronic liver disease. We separately analyzed these mechanisms in order to elucidate their specific role in PVT development.

### 4.1. Blood Stasis

Patients with chronic liver disease present many changes in portal circulation caused by histological alterations in the liver parenchyma: liver fibrosis drives the increase in the resistances in the portal venous system. At the same time, higher levels of vasodilators are released in the systemic bloodstream causing the vasodilatation of the splanchnic arterioles [81,82,83,84,85,86]. In this process, a redistribution of blood flow in the portal system is observed. In detail, the caliber of the portal vein enlarging and the opening of many portosystemic shunts occur in order to reduce portal hypertension and bypass the liver circulation [87]. Portal vein dimension and the presence of collaterals are risk factors of PVT development, since they cause a reduction in the blood flow and in portal blood velocity [86]. These mechanisms cause blood stasis that facilitates the adhesion and activation of platelets, contributing to the primary step of the hemostatic process [85]. Indeed, it has been demonstrated that blood flow velocity inferior to 15 cm/s is a predictor factor of PVT and might discriminate cirrhotic patients at higher risk of PVT development compared to patients with lower risk [87,88]. In this context, the administration of NSSBs remains controversial since they reduce portal resistances and stimulate splanchnic vasoconstriction [89]. Despite these concerns, no studies have demonstrated their role in worsening PVT.

### 4.2. Endothelial Dysfunction and Platelets Activation

Liver sinusoidal endothelial cells (LSECs) have specific characteristics such as fenestration and lack of a basal membrane, which favor a direct interaction between hepatocytes, sinusoids, and the blood nutrients and regulators. LSECs have a crucial role in the homeostasis of hepatocytes and are involved in many processes, including regulation of the vascular tone, hemostasis, and inflammation [90]. LSEC in healthy conditions produces several substances such as nitric oxide (NO), prostacyclin, and thrombomodulin which are involved in the integrity of the vessels and guarantee an adequate blood support and nutrients for hepatocytes. In the presence of chronic inflammation and endotoxemia, LSECs present morphological changes such as loss of fenestrae and expression of a basal membrane. Moreover, shear stress induces a direct damage of LSECs which causes the exposition of the extracellular matrix and the activation of subendothelial prothrombotic components and the loss of vasodilators described before. All together, these alterations lead to an increase in peripheral resistance which causes the worsening of portal hypertension [91,92,93,94,95]. The capillarization of LSECs influences the oxygenation of hepatocytes, resulting in an accelerated death process that hampers liver inflammation and activation of damage-associated molecular patterns (DAMPs) that stimulate hepatic stellate cells (HSCs) to produce extracellular matrix [96]. This process is exacerbated by a reduction in NO levels and upregulation of thromboxane A2, endothelin, and reactive oxygen species release [97]. In this condition, cytokines activate neutrophils, the Nuclear Factor kappa-light-chain-enhancer of activated B cells (NFkB) pathway, and the expression of P selectin, a regulator of endothelial-platelet attachment. They also upregulate intercellular adhesion molecule 1 (ICAM1), E selectin, and Interleukin 8 (IL-8) factors which contribute to amplify and maintain inflammation [98,99]. Moreover, in inflammatory conditions, LSECs store and produce vWF and Factor VIII (FVIII) which are pivotal for thrombotic pathways [90,98,99,100]. In particular, vWF is a multimeric glycoprotein released in multimers in the bloodstream [101]. It binds to platelets with an affinity which is proportional to its molecular weight. In normal conditions it is not expressed by LSECs, while in necrotic and congestive livers vWF is detected in the capillaries [102]. Its direct inhibitor is a disintegrin and metalloproteinase with a thrombospondin type 1 motif, member 13 (ADAMTS-13). It acts on vWF by cutting multimers between Tyr1605 and Met1606 in its A2 domain. ADAMTS-3 is stored in HSCs in hemostatic conditions. In the presence of chronic inflammation, ADAMTS-13 production is impaired due to the transformation of HSCs into myofibroblasts, while the capillarization of LSECs augments the production and release of high molecular weight vWF multimers with a greater ability in thrombi formation [103,104,105]. This imbalance between agonist and antagonist increases platelet deposition, thus causing the formation of platelet microthrombi and fibrin deposition in hepatic sinusoids with loss of liver parenchyma that, in a vicious cycle, activates inflammation and stimulates liver thrombosis [106,107,108]. Moreover, studies have demonstrated that HSCs stimulate the release of vascular endothelial growth factor (VEGF) and platelet-derived growth factor (PDGF) which upregulates angiogenesis in order to repair liver sinusoids, but during chronic inflammation this process is deregulated and contributes to portal hypertension maintenance since these new vessels are not capable to increase the blood perfusion of hepatocytes, worsening local ischemia [96,109]. Other mechanisms influence vWF levels, such as endotoxemia and gut dysbiosis: endotoxemia causes vWF release from endothelial cells and the upregulation of FVIII; moreover, an increased activation of tissue factor emerged in this condition with the consequent stimulation of Factor VII that leads to the activation of the coagulation cascade [110]. These mechanisms worsen endothelial damage with the release of inflammatory cytokines (IL 6), upregulating platelet activation, and microthrombi deposition. A direct effect of gut dysbiosis and endotoxemia were reported in the development of splanchnic vein thrombosis [111,112]. Platelet infiltration in liver tissue causes microthrombi deposition and cell apoptosis in a mechanism that amplifies local inflammation [113]. Platelets in cirrhotic patients express higher levels of isoprostanes that activate the glicoprotein (Gp) IIb/IIIa receptor inducing platelet aggregability. Moreover, platelets produce the transforming growth factor (TGF) beta that upregulated endothelial vWF, thrombomodulin (TM), ICAM-1, and VEGF, promoting endothelial dysfunction in in vitro experiments [114]. Another component of endothelial prothrombotic phenotype is related to microparticles derived from endothelial cells that express annexin V and cluster differentiation (CD) 62 mainly concentrated in the portal area. They act as a surface that facilitates blood aggregability. Moreover, extracellular vesicles expressing phosphatidylserine and tissue factor were reported to be higher in patients with advanced decompensated liver diseases and contribute to endothelial activation and thrombosis. Finally, extracellular vesicles derived from platelets expressing CD61 were reported to be higher in patients with hepatitis c virus (HCV) related cirrhosis and increased the risk of portal vein thrombosis [115].

### 4.3. Hypercoagulability

For decades, liver cirrhosis has been described as an acquired bleeding disorder due to thrombocytopenia and abnormal routine parameters of blood coagulation [116]. The latest evidence shows that in advanced chronic liver diseases a rebalance of the hemostasis occurs: despite low levels of platelets present, they express a pattern of hyperactivity. The increased levels of factors such as vWF and Factor VIII due to endothelial dysfunction and chronic inflammation are counterbalanced by the reduction in anticoagulant factors due to decreased synthesis of prothrombin, factor V and X, and anticoagulant factors (protein C, protein S, and antithrombin), as is expected due to reduced liver function [117,118,119,120,121]. Despite these changes, a perfect balance between procoagulant and anticoagulant factor is reported in patients with liver cirrhosis in thrombin generation (TG), detected by assays [120]. TG assay is a laboratory test that assesses the role of each procoagulant and anticoagulant factors on the global hemostatic status. They measure several parameters in a sample, such as lagtime to thrombin production, the time to thrombin peak, the peak thrombin levels, the endogenous thrombin potential (ETP), and the role of anticoagulant protein C or thrombomodulin (TM) [122]. Regarding the TG assay, differences in thrombin generation levels are reported according to different inclusive criteria in each patients’ selection or dissimilarities in test conditions. Lisman et at. observed that TG-TM tests were performed by adding different concentrations of tissue factor (TF), phospholipids, and rabbit thrombomodulin to the plasma. The results demonstrated that in patients with a MELD score below 12, and in the absence of TM, higher ETP was obtained compared to healthy subjects, while the addition of TF led to similar levels of thrombin in both groups. Moreover, the addition of protein C caused a reduction in ETP that was consistent in controls, with poor effects on cirrhotic patients, suggesting an acquired resistance in this class. Finally, patients with MELD up to 12 produced higher levels of thrombin at every TM and TF level. Despite these results, the difference in thrombin generation between the two groups did not reach statistical significance [123,124]. The characteristics of the test conditions may have influenced the results: in standard TG assays, platelet poor plasma and synthetic phospholipids are used to activate the coagulation cascade, and thus this does not reflect the physiological coagulation that occurs in the presence of whole blood. In one study, a whole blood thrombin generation assay was performed in 34 cirrhotic patients compared with 22 healthy donors. The cirrhotic patients demonstrated a longer lagtime and time to peak and lower peak thrombin values compared to controls. Despite these results, no difference in ETP was reported between cirrhotic and non-cirrhotic patients when the WB-TG assay was performed. Moreover, TG was more resistant to thrombomodulin in the patients compared to the controls in both whole blood and plasma, but the inhibitory effect of thrombomodulin was highly reduced in whole blood than in plasma. In this delicate balance, changes in homeostasis have been questioned as a cause of portal vein thrombosis development [125]. In this context, the ratio between FVIII and Protein C has been evaluated as a marker of rebalance hemostasis, reflecting the coagulation status as it increases with severity of liver disease. Its role in PVT development has been questioned, with controversial results. It is definitively correlated with fibrosis progression and risk of decompensation [117,118,119,120,121]. Other parameters of blood coagulation such as Factor II to Protein C ratio were evaluated as contributors of thrombosis development without positive results. Factors such as thrombomodulin, fibrinolysis markers, and plasminogen activator inhibitor 1 levels were evaluated in this context, but they did not predict PVT development. Taken together these results provide evidence that coagulation factors may not be a predictor of PVT but are more likely influenced by the severity of liver disease and portal hypertension. Decreased protein C and antithrombin levels have been associated with the risk of PVT, and these changes mainly occur in presence of decompensated liver disease. Among thrombophilic mutations, only Factor V Leiden and MTHFR were associated with higher risk of PVT in cirrhotic patients, with controversial results on the role of prothrombin G20210A. Antiphospholipid antibodies were reported in cirrhotic patients, with a prevalence which was directly influenced by the severity of liver disease; data on their role on PVT development are still lacking [126,127,128,129]. The role of fibrinolysis in this context was proven by several studies: patients with a clot lysis time above the 90th percentile had a 2-fold increased risk of venous thromboembolism, while high TAFI levels were associated with increased risk of thrombosis. PAI-1, t-PA, plasminogen and α2-antiplasmin did not show conclusive results [12,130]. Interestingly, the ratio between thrombin-antithrombin complex (TAT) and tissue plasminogen activator-inhibitor complex (tPAIC) seems to predict the development of PVT in cirrhotic patients [10].

## 5. Non-Cirrhotic Portal Vein Thrombosis

Thrombosis of the portal venous system rarely occurs in patients without a liver disease, and has a lifetime prevalence of 0.7–1% in the general population [131,132,133]. Sometimes PVT can be the first clinical manifestation of a tumoral disorder. In a retrospective cohort, patients with PVT received diagnosis of an intraabdominal neoplasia (mostly hepatic or pancreatic but also stomach and ovary) or hematologic malignancy [134]. Venous thrombotic events involving the splanchnic system occur at rates of approximately 5–10% for PV, 9–13% for ET, and 0.6–1% for PMF. PVT accounts for 40% of these cases. A large meta-analysis estimated that 30% of non-cirrhotic PVT patients have an underlying MPN [135]. Among other cancers, the revision of autopsies demonstrated that, within 254 patients with PVT, 28% of PVT was associated with primary or secondary hepatobiliary cancers, and pancreatic cancer patients had 67 times higher odds for PVT [133]. Risk factors in cancer-related thrombosis are related to patients’ characteristics and tumor characteristics. Older age, obesity, prolonged immobility, chemotherapies, smoking, and contraceptive drugs are well-known risk factors for the development of venous thromboembolism [133,136]. Cancer cells are able to induce inflammatory cytokines which upregulate the expression of tissue factors and other procoagulant factors and may induce its production in endothelial cells and in monocytes and macrophages. This exaggerated release of Tissue factor causes the activation of Factor VII, leading to the activation of the extrinsic pathway of the coagulation cascade [136,137]. Cancer cells are also able to induce the activation of factor X via a cysteine protease, acting on the intrinsic pathway of coagulation. Pancreatic and ovary tumors also may activate L and P selectins through platelets and endothelial activation and thrombus formation [138]. Among hematologic malignancies, the role of myeloproliferative disorders is well established in PVT and, interestingly, PVT may be the first sign of MPNs. MPNs may cause both arterial and venous thrombosis (PV: 28.6%; ET: 20.7%; PMF: 9.5% at time of diagnosis). This may be explained since MPN patients have elevated levels of cell free hemoglobin and nitric oxide and this may be a cause of oxidative stress. Moreover, they express higher levels of plasminogen activator inhibitor 1, with higher risk of hypofibrinolisis. The presence of JAK2 mutation (JAK2V617F) is strictly associated with PVT (OR up to 50) [139]. JAK2 mutations induce a prothrombotic and inflammatory state, not only enhancing blood viscosity due to blood cell proliferation, but also inducing a procoagulant phenotype in platelets and endothelial cells that express a higher tendency to aggregation and higher adherence to the endothelium. JAK2 is also expressed in the endothelium of the liver and spleen, where it induces higher levels of P selectin, causing similar effects on platelet aggregation [140,141,142,143,144,145,146,147]. The molecular mechanisms related to venous thrombosis induced by MPNs are discussed in Section 1 of this review. Non-tumoral causes of non-cirrhotic PVT include abdominal inflammation (pancreatitis and cholecystitis predominantly), sepsis, abdominal trauma, and surgery. Inflammation is a main driver of thrombosis due to the release of cytokines able to activate neutrophils, platelets, and the endothelial cells with a prothrombotic phenotype; bacteremia has the same effects [148]. A study demonstrated that *Bacteroidetes* may favor thrombosis, thereby inducing the appearance of anticardiolipin antibodies [149]. Sepsis derived by intraabdominal infections may cause non-cirrhotic PVT, namely pylephlebitis: pathogens spread through local venules and veins that drain into the portal venous system. *E. coli* was the most common pathogen detected in blood cultures, the other pathogens were *Streptococcus* spp. and *Bacteroides* spp. [150]. In a metanalysis, pylephlebitis was caused by a single pathogen in 42.8% cases, whereas it was polymicrobial in 27.2% cases. The responsible pathogen was not detected in 30% of the included population [151]. Post-traumatic PVT due to injury of the abdomen, such as sport related trauma, is another cause of PVT in non-cirrhotic patients [152].

In patients without cirrhosis, the main causes of thrombosis are linked to thrombophilic disorders or myeloproliferative malignancy [139,153]. In this context, the leading cause of thrombosis is linked to thrombophilic mutations. Factor V Leiden mutation (8%) and prothrombin G20210A mutation (2%) have the higher prevalence in the general population, while deficiency of protein S, protein C, or antithrombin has a low prevalence of <1% in this group. Between these, factor V Leiden mutation and protein C deficiency have been associated with PVT in a study by Janssen et al. comparing patients with and without PVT [153]. Moreover, in a meta-analysis the presence of factor V Leiden and prothrombin G20210A mutations in PVT patients is significantly described [154]. The methylenetetrahydrofolate reductase C677T and plasminogen activator inhibitor—type 1 4G–4G mutations were also described as independent predictors of PVT but studies on their role on thrombotic risk are controversial [155]. Presence of antiphospholipid (aPL) antibodies is the most important acquired thrombophilia with a prevalence of 1–16%. In particular, anticardiolipin antibodies increase the risk of PVT in both cirrhotic and non-cirrhotic patients [156,157].

## 6. Discussion and Future Perspectives: The Role of Anticoagulation

Vascular disorders of the liver comprise a series of diseases with a high risk of portal hypertension development, and are characterized by an increased risk of thrombosis. Studies on VLD are hampered by its low prevalence in the general population and by the lack of animal models that may reflect the characteristics of the patients, leading to a gap of knowledge in the management of these diseases. In normal conditions, the microvasculature of the liver, the sinusoids, regulates the homeostasis and is able to maintain a stable blood supply for the nourishment of hepatic stellate cells and hepatocytes. Development of portal hypertension is the first step to determining dramatic changes in the liver lobule: blood flow reduction and capillarization of endothelial cells, favoring a hypercoagulable environment (Figure 1). Based on this evidence, the role of anticoagulation needs to be elucidated in the management of VLD. According to Baveno VII’s statements [66], anticoagulation is recommended in cirrhotic patients with recent PVT of the trunk with or without involvement of the mesenteric vein, in symptomatic patients with any thrombus extension, and in patients on the waiting list for LT regardless of the grade of thrombosis. Anticoagulation should be evaluated in patients with thrombosis progression during short term follow up of 1–3 months and in case of the involvement of the superior mesenteric vein. The duration of anticoagulation should be maintained for a minimum of 6 months but it may be prolonged until the complete recanalization; furthermore, in patients who are candidates for liver transplantation it is possible to continue treatment until surgery. In patients with PVT without cirrhosis anticoagulation should be started immediately and a long-term duration is recommended in patients with permanent pro-thrombotic condition [66]. A recent randomized trial demonstrated that Rivaroxaban reduced the incidence of venous thromboembolism in patients with chronic non-cirrhotic PVT without exhibiting major risks for thrombosis [158]. Likewise, long-term anticoagulation should be given to patients with BCS. No recommendation can be made about the use of anticoagulation to prevent PVT in PSVD, and this remains an unmet need [66]. Currently, whether PVT occurrence leads to the worsening of natural history of chronic liver disease is debated; however, on the other hand, as previously described, growing evidence suggests a direct role of anticoagulants in the prevention of liver fibrosis and parenchymal extinction acting on microthrombosis with a reduction in hepatic decompensation risk [107,159,160,161,162,163,164,165]. Indeed, a study conducted in rats with carbon tetrachloride (CCL4) induced liver damage demonstrated a reduction in fibrosis after low weight molecular heparin (LWMH) administration associated with an improvement in liver regeneration [166]. Enoxaparin administration lowered the grade of parenchymal necrosis and fibrosis as well as ameliorated hepatocyte necrosis indices in a model of mice in which parenchymal necrosis was provoked by the ligation of the bile ducts [167]. Moreover, in two models of cirrhosis in rats HSCs (CCl4 induced and thioacetamide induced), the administration of enoxaparin lowered portal pressure and hepatic vascular resistance and reduced the presence of reactive oxygen species [168]. In this context, long-term therapy with enoxaparin was associated with a significant reduction in hepatic venule resistance, portal pressure, and liver tissue fibrosis compared to untreated controls and with an improvement in microthrombosis [169]. Based on these observations, the role of anticoagulants in cirrhotic patients has gained interest as a therapeutic choice to reduce the risk of liver failure and death for all hepatic causes [169,170,171]. Preclinical studies on DOAC in cirrhosis have demonstrated beneficial effects on overall survival and fibrosis reduction: rivaroxaban and dabigatran reduced fibrosis in two different models of cirrhosis. In detail, rats treated with CCl4 were randomized into three groups: one group received rivaroxaban 5 mg/kg versus rats, one went without therapy, and formed the controls. Rats in the rivaroxaban group showed reduced fibrosis markers, tissue factor, fibrin, and alpha smooth muscle actin levels in liver tissue, suggesting a role in attenuating CCL4-induced liver damage. In another study, animals receiving rivaroxaban 20 mg/kg/day achieved a significant reduction in portal pressure and vascular resistances. Reduction in portal pressure and hepatic vascular resistances and a lower expression of profibrotic factors (α-smooth muscle actin, collagen type I alpha 1 chain, tissue inhibitor metallopeptidase ½, PDGF beta, and TGFβ) were observed. Rats treated with rivaroxaban also expressed a reduction in vWF and a better response to acetylcholine stimulation, suggesting an improvement in endothelial functions and a reduction in HSCs activation [172,173]. Nowadays, a beneficial effect of anticoagulation emerged also in cirrhotic patients and, interestingly, this effect seems disconnected from the presence of PVT [174,175,176,177,178,179,180,181]. Recently, a large meta-analysis included 500 patients from five studies with non-tumoral portal vein thrombosis evaluated the role of anticoagulation on all causes of mortality, bleeding events, and PV recanalization. All the studies compared anticoagulation (LMWH and/or vitamin K antagonists) with no treatment with a median duration of 9 months and a median follow up of 11.7 to 44 months. In total, 205 patients received anticoagulation. Patients who did not receive treatment had higher Child Pugh and Model for End Stage Disease (MELD) scores, while patients in the anticoagulation group had a higher rate of beta blocker therapy. Results showed that in a total 161 patients died, 24.9% were in the anticoagulation group compared to 39.1% of patients in the no treatment group. Liver-related mortality was also lower in the anticoagulation group (9.3 vs. 20.3%). Death by all causes was markedly reduced by anticoagulant therapy and its effects remained statistically significant after the adjustment for liver function scores. The effect of anticoagulation on all-cause mortality remained similar regardless of severity or rate of recanalization of portal vein thrombosis, and the risk of death decreased in patients with a long duration of anticoagulation. No differences in terms of bleeding were observed between the two groups [182]. The preliminary results of a randomized multicenter study comparing rivaroxaban with no treatment in cirrhotic patients reported positive effects on survival in anticoagulated patients, including Child Pugh B ones [183]. Taken together, the beneficial effects of anticoagulation exceed the effects on portal recanalization, thus ameliorating overall survival. Further studies are needed to provide details on the role of anticoagulation on liver fibrosis and the effects on outcome and survival in patients with VLD.

## 7. Conclusions

Vascular disorders of the liver are a wide group of rare diseases affecting patients that are frequently younger and healthier compared to other patients with chronic liver disease. The development of portal hypertension is a crucial step in the progression of these diseases. Thrombosis may be described as one of the complications related to portal hypertension. The role of blood stasis, the alteration of the endothelial barrier, and the alteration in coagulation lead to a prothrombotic state, resulting in the rebalance of the hemostasis. Thrombosis increases, in a vicious circle, portal hypertension, thereby favoring the risk of liver decompensation. Other factors such as thrombophilia, tumors, and congenital or acquired conditions have the same results on liver vasculature. In this context, studies to better understand this molecular pathogenesis are fundamental to identifying how to manage these diseases in order to improve the outcome of patients. The role of anticoagulation is promising in this context. Patients with PVT who are treated with anticoagulants present a reduced risk of hepatic decompensation even in the absence of recanalization and tend to have better outcomes in terms of overall survival. Similar effects have also been historically reported in patients with BCS. Conversely, no final consensus has been obtained in the use of anticoagulation to prevent PVT in patients with PSVD. Large randomized clinical trials are required to assess the benefit risk ratio of prophylactic anticoagulation in this setting.

## Figures and Tables

**Figure 1 ijms-24-12754-f001:**
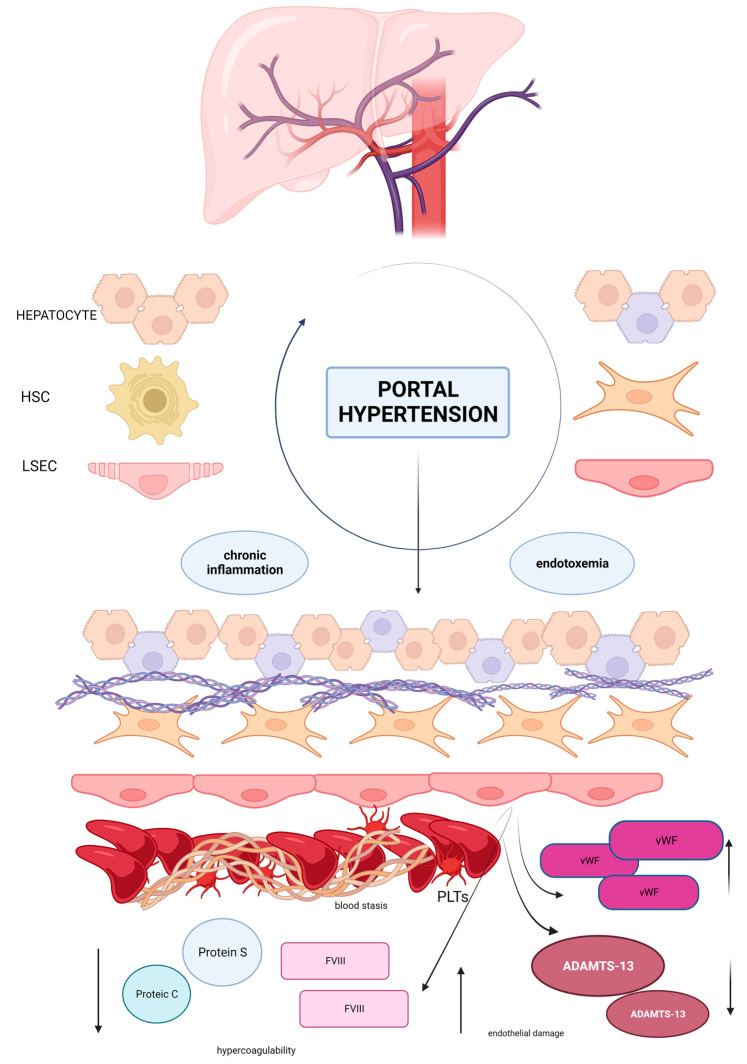
The healthy sinusoid is composed by a monolayer of endothelial fenestrated cells able to regulate blood homeostasis with the production of nitric oxide and other similar molecules that maintain the perfect blood supply for hepatic stellate cells and hepatocytes. In presence of portal hypertension blood flow is reduced and, as an effect, blood stasis induces a hypercoagulable state. Endothelial cells lose their fenestrae and express prothrombotic factors such as Factor VIII and vWF which aggregate in large multimeters, favoring platelet aggregability. Hepatic production of anticoagulants is impaired. A low grade of chronic inflammation and endotoxemia influence this pathway, enhancing the prothrombotic environment. Hepatic stellate cells lose the ability to produce anticoagulants such as ADAMTS-13, the inhibitor of vWF, leading to an imbalance between prothrombotic factors and anticoagulants which causes thrombi deposition in liver sinusoids. Moreover, activated hepatic stellate cells induce the production of collagen in the interstitial space, resulting in fibrogenesis and death of hepatocytes. This model perfectly describes the role of Virchow’s triad in thrombosis development and the link between thrombosis and hepatic fibrogenesis. Abbreviations: HSC hepatic stellate cells, LSEC liver sinusoidal endothelial cells, FVIII Factor VIII, vWF von Willebrand factor, PLTs platelets, ↑ increased, ↓ decreased.

## Data Availability

Not applicable.

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
