# Peer review of "Molecular Mechanisms Underlying Vascular Liver Diseases: Focus on Thrombosis"

_ijms, 2023, doi:10.3390/ijms241612754_

Round 1

Reviewer 1 Report

I have read with interest this well written paper on vascular liver diseases. The work is interesting, pertinent for clinicians and researchers and in my opinion contributes to the current literature on this topic. This being said, I do have several major remarks that I would kindly ask authors to address: 

1) Line 60- thrombosis associated with celiac disease and inflammatory bowel disease is missing references in this section: Please read the following newer publications : 

Celiac Disease and Thrombotic Events: Systematic Review of Published Cases - PubMed (nih.gov)

Thromboembolic Events in Patients with Inflammatory Bowel Disease: A Comprehensive Overview - PubMed (nih.gov)

2) Line 61-74- I think this part is too detailed, takes focus away from BDS- in my opinion it would be enough just to list MPNs without diving into too many details.

3) Non cirrhotic PVT- the authors have not mentioned two entities:

sport related portal vein thrombosis (usually after blunt traumatic injury) and

septic portal vein thrombosis (pylephlebitis). These should be added in this paragraph: please see the following: 

Suppurative Thrombosis of the Portal Vein (Pylephlebits): A Systematic Review of Literature - PubMed (nih.gov)

Pylephlebitis: A Systematic Review on Etiology, Diagnosis, and Treatment of Infective Portal Vein Thrombosis - PubMed (nih.gov)

References must be updated with newer publications and some obsolete ones should be removed. In general references should be within 10 years, maximum 15 years of publications. The authors have references from 1985- ref 134, for example. 

other that seem obsolete are 6, 7, 21, 44, 48, 51. 53, 55, 56, 57, 60, 68, 79, 89, 94, 99, 103-106

some grammatical errors, sentence structure etc

Author Response

1) Line 60- thrombosis associated with celiac disease and inflammatory bowel disease is missing references in this section: Please read the following newer publications :

Celiac Disease and Thrombotic Events: Systematic Review of Published Cases - PubMed (nih.gov)

Thromboembolic Events in Patients with Inflammatory Bowel Disease: A Comprehensive Overview - PubMed (nih.gov)

​​We thank the Reviewer for the suggestion, we have added these two references (page 2, line 60)

2) Line 61-74- I think this part is too detailed, takes focus away from BDS- in my opinion it would be enough just to list MPNs without diving into too many details.

We thank the Reviewer for the suggestion, we have modified the text accordingly (page 2, line 66-70).

3) Non cirrhotic PVT- the authors have not mentioned two entities: sport related portal vein thrombosis (usually after blunt traumatic injury) and septic portal vein thrombosis (pylephlebitis). These should be added in this paragraph: please see the following:

Suppurative Thrombosis of the Portal Vein (Pylephlebits): A Systematic Review of Literature - PubMed (nih.gov)

Pylephlebitis: A Systematic Review on Etiology, Diagnosis, and Treatment of Infective Portal Vein Thrombosis - PubMed (nih.gov)

We thank the Reviewer for the suggestions, we have included them in the manuscript (page 9, lines 419-428)

4) References must be updated with newer publications and some obsolete ones should be removed. In general references should be within 10 years, maximum 15 years of publications. The authors have references from 1985- ref 134, for example, other that seem obsolete are 6, 7, 21, 44, 48, 51, 53, 55, 56, 57, 60, 68, 79, 89, 94, 99, 103-106

We thank the Reviewer for the comment, we have updated some of the references and removed some obsolete ones. We maintained the references 55-56 (now 59-60) related to the first animal studies on non-cirrhotic portal hypertension and references 103-105 because we think they are useful for the comprehension of ADAMTS-13 characteristics.

Reviewer 2 Report

In this manuscript, Giuli et al provided a detailed overview of the current knowledge of molecular mechanisms underlying VLD, and highlighted promising role of anticoagulation for preventing liver complications and to improve the outcome of these patients.

Comments:

# Ref#106 seems to be numbered incorrectly, there are two refs numbered 106.

# line 324-350, the authors discussed the effect of hypercoagulability on VLD. However, thrombomodulin (TM) modified thrombin generation (TG) was not always comparable between liver cirrhotic patients and healthy controls: Lisman repeated reported enhanced TM-modified TG in cirrhotic patients in PPP (PMC6058283; PMID: 25065556; PMC7186949). Interestingly, comparable TM-modified TG in cirrhotic patients when tested in whole blood, suggesting assay conditions can influence the comparisons between cirrhotic patients and healthy controls. The authors are suggested to include these discussions in their review.

# could the author also briefly discuss the state of fibrinolysis in VLD?

Author Response

1) Ref#106 seems to be numbered incorrectly, there are two refs numbered 106.

We thank the Reviewer, we have changed it.

2)  line 324-350, the authors discussed the effect of hypercoagulability on VLD. However, thrombomodulin (TM) modified thrombin generation (TG) was not always comparable between liver cirrhotic patients and healthy controls: Lisman repeated reported enhanced TM-modified TG in cirrhotic patients in PPP (PMC6058283; PMID: 25065556; PMC7186949). Interestingly, comparable TM-modified TG in cirrhotic patients when tested in whole blood, suggesting assay conditions can influence the comparisons between cirrhotic patients and healthy controls. The authors are suggested to include these discussions in their review.

This is an interesting topic and as You suggest, we add a discussion about it (page 7, lines 328-357)

3) could the author also briefly discuss the state of fibrinolysis in VLD?

We added a brief paragraph regarding BCS and fibrinolysis at page 2, lines 60-65, and a paragraph on PVT at page 8, lines 373-379

Round 2

Reviewer 1 Report

I would like to thank the author for careful revisions

n/a